# A note on regularised NTK dynamics with an application to PAC-Bayesian training

**Eugenio Clerico***  *eugenio.clerico@gmail.com*
*Universitat Pompeu Fabra, Barcelona*

**Benjamin Guedj**  *b.guedj@ucl.ac.uk*
*Centre for AI and Department of Computer Science, University College London & Inria London*

**Reviewed on OpenReview:** *https://openreview.net/forum?id=2la55BeWwy*

## Abstract

We establish explicit dynamics for neural networks whose training objective has a regularising term that constrains the parameters to remain close to their initial value. This keeps the network in a *lazy training* regime, where the dynamics can be linearised around the initialisation. The standard neural tangent kernel (NTK) governs the evolution during the training in the infinite-width limit, although the regularisation yields an additional term that appears in the differential equation describing the dynamics. This setting provides an appropriate framework to study the evolution of wide networks trained to optimise generalisation objectives such as PAC-Bayes bounds, and hence contribute to a deeper theoretical understanding of such networks.

## 1 Introduction

The analysis of infinitely wide neural networks can be traced back to Neal (1995), who considered this limit for a shallow (1-hidden-layer) network and showed that, before the training, it behaves as a Gaussian process when its parameters are initialised as independent (suitably scaled) normal distributions. A similar behaviour was later established for deep architectures, also allowing for the presence of skip-connections, convolutional layers, *etc.* (Lee et al., 2018; 2019; Arora et al., 2019a; Novak et al., 2019; Garriga-Alonso et al., 2019; Yang, 2019; Hayou et al., 2021). Lee et al. (2019; 2020), among others, brought empirical evidence that wide (but finite-size) architectures are still well approximated by the Gaussian limit, while finite size corrections were derived in Antognini (2019) and Basteri & Trevisan (2022).

Although the previous results hold only at the initialisation (as the Gaussian process approximation is only valid before the training), Jacot et al. (2018) established that the evolution of an infinitely wide network can still be tracked analytically during the training, under the so-called neural tangent kernel (NTK) regime. In a nutshell, they showed that the usual gradient flow on the parameters space induces the network's output to follow a kernel gradient flow in functional space, governed by the NTK. A main finding of Jacot et al. (2018) is that although for general finite-sized networks the NTK is random at the initialisation and evolves during the training, in the infinite-width limit it becomes a deterministic object that can be exactly computed, and it stays fixed throughout the training. Later, Lee et al. (2019) provided a new proof of the convergence to the NTK regime, while Yang (2019) established similar results for more general architectures, such as convolutional networks. Chizat et al. (2019) extended the idea of linearised dynamics to more general models, introducing the concept of *lazy training* and finding sufficient conditions for a network to reach such regime. They also pointed out that this linearised behaviour may be detrimental to learning, as also highlighted by Yang & Hu (2022) who showed that the NTK dynamics prevent a network hidden layers from effectively learning features. However, the NTK has been a fruitful tool to analyse convergence (Allen-Zhu et al.,

---

*Work mostly done while affiliated at the Department of Statistics, University of Oxford, UK.

2019b; Du et al., 2019) and generalisation (Allen-Zhu et al., 2019a; Arora et al., 2019b; Cao & Gu, 2019) for over-parameterised settings under (stochastic) gradient descent.

The standard derivation of the NTK dynamics (Jacot et al., 2018; Lee et al., 2019) requires the network to be trained by gradient descent to optimise an objective that depends on the parameters only through the network's output. This setting does not allow for the presence of regularising terms that directly involve the parameters. Yet, in practice, a network reaches the NTK regime when its training dynamics can be linearised around the initialisation. This happens if the network parameters stay close enough to their initial value throughout the training, the defining property of the *lazy training* regime. It is then natural to expect that a regularising term that enforces the parameters to stay close to their initialisation will still favour linearised dynamics, and so bring a training evolution that still can be expressed in terms of a fixed and deterministic NTK. This is the focus of the present paper, where we discuss the evolution of a network in the NTK regime trained with an $\ell^2$-regularisation that constrains the parameters to stay close to their initial values. We remark that similar ideas are also present in Hu et al. (2020), where the authors study the evolution of a linearised approximation of a neural network under the $\ell^2$-regularisation, without however proving the convergence of the original network's dynamics to those of the linearised model.

We note that regularisers centred at the initialisation typically appear in PAC-Bayes-inspired training objectives, where the mean vector of normally distributed stochastic parameters is trained via (stochastic) gradient descent on a generalisation bound (an approach initiated by the seminal work of Langford & Caruana, 2001 and further explored by Alquier et al., 2016; Dziugaite & Roy, 2017; Neyshabur et al., 2018; Letarte et al., 2019; Nagarajan & Kolter, 2019; Zhou et al., 2019; Nozawa et al., 2020; Biggs & Guedj, 2021; 2022; Dziugaite et al., 2021; Pérez-Ortiz et al., 2021a;b; Pérez-Ortiz et al., 2021; Chérief-Abdellatif et al., 2022; Lotfi et al., 2022; Tinsi & Dalalyan, 2022; Clerico et al., 2022; 2023a; Viallard et al., 2023). As these training objectives yield generalisation guarantees, we conjecture that the exact dynamics that we derive could be a starting point to obtain generalisation bounds for more general kernel gradient descent algorithms.

As final remarks, we note that Chen et al. (2020) considers a NTK regime that allows for regularisation, but they only consider the mean-field setting of two-layer neural networks (introduced by Chizat & Bach, 2018; Mei et al., 2018, and further explored by Mei et al., 2019; Wei et al., 2019; Fang et al., 2019), where the network is not initialised with a scaling yielding a Gaussian process. Finally, we mention that also Huang et al. (2022) attempted the analysis of PAC-Bayesian dynamics via NTK. However, their approach differs from ours, it does not highlight the effect of the regularisation term, and the whole analysis deals with a simple shallow stochastic architecture where only one layer is trained.

**Outline.** We present our framework and notation in Section 2 and then treat the unregularised NTK dynamics in Section 3 as a starter. We then move on to the more interesting case of regularised dynamics in Section 4, first for the simple case of $\ell^2$-regularisation and then for a more general regularising term. We instantiate our analysis to the example of least square regression in Section 5 and illustrate the merits of our work with an application to PAC-Bayes training of neural networks in Section 6. The paper ends with concluding remarks in Section 7 and we defer technical proofs to Appendix A.

## 2 Setting and notation

We consider a fully-connected feed-forward neural network of depth $L$, which we denote as $F : \mathcal{X} \to \mathbb{R}^q$, where $\mathcal{X} \subset \mathbb{R}^p$ is a compact set. We denote as $n_l$ the width of the $l$-th hidden layer of the network, as $n_0 = p$ the input dimension and $n_L = q$ the output dimension. We consider the network

$$F(x) = U^L(x); \qquad U_i^{l+1}(x) = \frac{1}{\sqrt{n_l}} \sum_{j=1}^{n_l} W_{ij}^{l+1} \phi(U_j^l(x)); \qquad U_i^1(x) = \frac{1}{\sqrt{n_0}} \sum_{j=1}^{n_0} W_{ij}^1 x_j;$$

where $\phi$ is the network's activation, acting component-wise. The network's prediction in the label space $\mathcal{Y}$ is $\hat{y}(x) = f(F(x))$, for some $f : \mathbb{R}^q \to \mathcal{Y}$. We denote as $\mathcal{W}$ the parameter space where the weights lie, and

as $W$ the parameters of the network. We consider the infinite-width limit, where all the hidden widths $n_l$ ($l = 1, \ldots, L-1$) are taken to infinity[1].

Data consists of pairs instance-label $z = (x, y) \in \mathcal{Z} = \mathcal{X} \times \mathcal{Y}$, with $x \in \mathcal{X}$ and $y \in \mathcal{Y}$. We consider a non-negative loss function $\ell : \mathcal{W} \times \mathcal{Z} \to [0, \infty)$. For a dataset $s \in \mathcal{Z}^m$, we define the empirical loss $\mathcal{L}_s$ as the average of $\ell$ on $s$, namely

$$\mathcal{L}_s(W) = \frac{1}{m} \sum_{z \in s} \ell(W, z) \,.$$

As we will often encounter empirical averages, we define the following handy notation

$$\langle g(Z) \rangle_s = \langle g(X, Y) \rangle_s = \frac{1}{m} \sum_{(x,y) \in s} g(x, y) \,,$$

so that $\mathcal{L}_s(W) = \langle \ell(W, Z) \rangle_s$. We assume that $\ell$ depends on $W$ only through the network's output $F$, *i.e.*, there exists a function $\hat{\ell}$ such that we can rewrite

$$\ell(W, z) = \hat{\ell}(F(x), y) \,.$$

The network training follows the gradient of a learning objective $\mathcal{C}_s$, namely

$$\partial_t W(t) = -\nabla \mathcal{C}_s(W(t)) \,,$$

where $\nabla$ denotes the gradient with respect to the parameters. We assume that $\mathcal{C}_s$ can be split into two terms, the empirical loss, and a regularisation term that depends directly on the parameters (without passing through the network's output $F$) and does not depend on $s$:

$$\mathcal{C}_s = \mathcal{L}_s + \lambda \mathcal{R} \,, \tag{1}$$

for some $\lambda \geq 0$. We let $\rho : \mathbb{R}^+ \to \mathbb{R}^+$ be a strictly increasing differentiable function such that $\rho(0) = 0$, and we consider the case of a regulariser in the form

$$\mathcal{R}(W) = \rho \left( \frac{1}{2} \|W - W(0)\|_{\mathrm{F}}^2 \right) \,, \tag{2}$$

with $W(0)$ denoting the value of the parameters at the initialisation, and with $\| \cdot \|_{\mathrm{F}}^2$ denoting the square of the Frobenius norm on $W$, namely $\|W\|_{\mathrm{F}}^2 = \sum_{l,i,j} (W_{ij}^l)^2$.

For conciseness, we write $\Delta g(t)$ for $g(t) - g(0)$, where $g$ is any time-dependent term. Moreover, we introduce the following notation

$$\psi_{k;ij}^{l;l'}(x; t) = \frac{\partial U_k^l(x; t)}{\partial W_{ij}^{l'}} \,;$$

$$\Theta_{kk'}^l(x, x'; t) = \sum_{l'=1}^{l} \psi_k^{l;l'}(x; t) \cdot \psi_{k'}^{l;l'}(x'; t) \,; \tag{3}$$

$$\Xi_k^l(x; t) = \sum_{l'=1}^{l} \psi_k^{l;l'}(x; t) \cdot \Delta W^{l'}(t) \,,$$

where '$\cdot$' denotes the component-wise inner product between matrices (or vectors) of the same size. $\Theta^L$ is the so-called neural tangent kernel (NTK) of the network. We remark that if $\Xi^l$ represent the linear approximation of the variation of $U^l$ with respect to changes in the parameters.

---

[1]Following the approach of Jacot et al. (2018), we will consider the case where this limit is taken recursively, layer by layer (that is we also have $n_{l+1}/n_l \to 0$).

## 3 The unregularised dynamics

Without regularisation (namely when we set $\rho \equiv 0$) the standard NTK dynamics hold (Jacot et al., 2018; Lee et al., 2019). We briefly cover this case, where the width of the network is taken to infinity.

At the initialisation, the network's output behaves as a centred Gaussian process, whose covariance kernel can be computed recursively. More precisely, we have that all the components of the output are i.i.d. Gaussian processes defined by

$$F_k \sim \mathcal{GP}(0, \Sigma^L),$$

where

$$\Sigma^1(x, x') = \frac{x \cdot x'}{n_0}; \qquad \Sigma^{l+1}(x, x') = \mathbb{E}_{(\zeta,\zeta') \sim \mathcal{N}(0, \Sigma^l(x,x'))}[\phi(\zeta)\phi(\zeta')]. \tag{4}$$

Moreover, the neural tangent kernel $\Theta^L$, defined in (3), tends to a diagonal deterministic limit (with respect to the initialisation randomness), and stays constant during the training. In particular, we have

$$\Theta^L_{kk'}(x, x'; t) = \bar{\Theta}^L(x, x')\delta_{kk'},$$

where $\bar{\Theta}^L$ can be computed recursively as follows:

$$\bar{\Theta}^1 = \Sigma^1; \qquad \bar{\Theta}^{l+1}(x, x') = \Sigma^{l+1}(x, x') + \mathbb{E}_{(\zeta,\zeta') \sim \mathcal{N}(0, \Sigma^l(x,x'))}[\dot{\phi}(\zeta)\dot{\phi}(\zeta')]\bar{\Theta}^l(x, x'), \tag{5}$$

with $\dot{\phi}$ denoting the derivative of $\phi$.

During the training, the network's output obeys the dynamics

$$\partial_t F_k(x; t) = -\frac{1}{m} \sum_{(\bar{x},\bar{y}) \in s} \bar{\Theta}^L(x, \bar{x}) \frac{\partial \hat{\ell}(F(\bar{x}; t), \bar{y})}{\partial F_k} = -\left\langle \bar{\Theta}^L(x, X) \frac{\partial \hat{\ell}(F(X), Y)}{\partial F_k} \right\rangle_s. \tag{6}$$

We remark that the network being in the NTK regime simply means that the model is linear around the initialisation. Indeed, neglecting terms of order $O(\|\Delta W(t)\|^2)$ one has

$$F(x; t) \simeq F(x; 0) + \mathrm{J}_W[F(X; 0)] \Delta W(t),$$

with $\mathrm{J}_W$ denoting the Jacobian with respect to the parameters. Assuming that this linear approximation holds we also have

$$\partial_t F(x; t) = \mathrm{J}_W[F(X; 0)] \partial_t W(t).$$

Now, up to terms of order $O(\|\Delta W(t)\|)$,

$$\partial_t W(t) = -\nabla \mathcal{L}_s(W(t)) = -\left\langle \mathrm{J}_W[F(X; t)]^\top \nabla_F \hat{\ell}(F(X; t), Y) \right\rangle_s \simeq -\left\langle \mathrm{J}_W F(X; t)^\top \nabla_F \hat{\ell}(F(X; 0), Y) \right\rangle_s.$$

Jacot et al. (2018) showed that in the infinite-width limit this linear approximation becomes exact, and $\mathrm{J}_W[F(x'; 0)] \mathrm{J}_W[F(X; 0)]^\top$ tends in probability to the $\bar{\Theta}(x, x') \mathrm{Id}$, leading to the dynamics (6), as

$$\partial_t F(x; t) \simeq -\mathrm{J}_W[F(X; 0)] \left\langle \mathrm{J}_W[F(X; t)]^\top \nabla_F \hat{\ell}(F(X; t), Y) \right\rangle_s \simeq -\left\langle \bar{\Theta}^L(x, X) \nabla_F \hat{\ell}(F(X), Y) \right\rangle_s.$$

## 4 The regularised dynamics

In this section, we discuss the impact of the regularisation term on the NTK dynamics. We first particularise this to the specific case of $\ell^2$-regularisation, then move to the treatment of more general regularisers.

Intuitively, we note that the regulariser that we are studying tends to keep the walues of the parameters close to their initialisation. Hence, we still expect that the dynamics can be linearised. If this assumption indeed holds, we get (in the simpler case $\rho = \mathrm{id}$)

$$\partial_t W_t \simeq -\left\langle \mathrm{J}_W[F(X; 0)]^\top \nabla_F \hat{\ell}(F(X; 0), Y) \right\rangle_s - \lambda \Delta W(t).$$

Keeping in mind that for the linearised model we have $\Delta F(x;t) \simeq \mathrm{J}_W[F(X;0)]\,\Delta W(t)$, now we have

$$\partial_t F(x;t) = -\left\langle \bar\Theta^L(x,X)\nabla_F \hat\ell(F(X),Y)\right\rangle_s - \lambda\Delta F(x;t)\,,$$

which is a regularised version of the NTK evolution. We will establish this more rigorously in the next sections, showing that under regularised dynamics the linearised model is a valid approximation of the neural network in the infinite-width limit.

### 4.1 Simple $\ell^2$-regularisation

We first consider the case $\rho \equiv \mathrm{id}$ in (2), so that $\mathcal{R}(W) = \frac{1}{2}\|\Delta W\|_{\mathrm{F}}^2$ and

$$\mathcal{C}_s(W) = \mathcal{L}_s(W) + \frac{\lambda}{2}\|\Delta W\|_{\mathrm{F}}^2\,. \tag{7}$$

By just applying the chain rule to $\partial_t W(t) = -\nabla\mathcal{C}_s(W(t))$, we find that

$$\partial_t W_{ij}^l(t) = -\nabla\mathcal{C}_s(W(t)) = -\sum_{k=1}^{n_L}\left\langle \psi_{k;ij}^{L;l}(X;t)\frac{\partial\hat\ell(F(X;t),Y)}{\partial F_k}\right\rangle_s - \lambda\Delta W_{ij}^l(t)\,. \tag{8}$$

This translates into the following functional evolution of the network's output

$$\partial_t F_k(x;t) = -\sum_{k'=1}^q\left\langle \Theta_{kk'}^L(x,X;t)\frac{\partial\hat\ell(F(X;t),Y)}{\partial F_{k'}}\right\rangle_s - \lambda\Xi_k^L(x;t)\,.$$

Our goal is to prove that in the infinite width limit the next two properties hold:

1. The NTK is constant at its initial value, which coincides with the standard deterministic NTK in (5), and more generally for all layers

$$\Theta_{kk'}^l(x,x';t) = \delta_{kk'}\bar\Theta^l(x,x')\,;$$

2. The term $\Xi^L$ is exactly $\Delta F$, and more generally

$$\Xi^l(x;t) = \Delta U^l(x;t) = U^l(x;t) - U^l(x;0)\,.$$

We note that these two properties are actually rather intuitive: the first one tells us that the Jacobian of the output stays fixed during the training, as if the dynamics where linear; the second one that we can indeed linearise $U$ around the initialisation. We recall that the standard NTK regime (with no regularisation) holds when the parameters do not move too much from their initial values, so that the dynamics can be linearised. The addition of a regularising term does actually enforce the parameters to stay close to their initial value. Hence, adding regularisation does not hinder the linearisation of the learning dynamics.

From the two properties above, we conclude that the NTK evolution of $F$ is given by the following.

**Theorem 1.** *In the infinite-width limit, taken recursively layer by layer, the network's output evolves as*

$$\partial_t F_k(x;t) = -\frac{1}{m}\sum_{(\bar x,\bar y)\in s}\bar\Theta^L(x,\bar x)\frac{\partial\hat\ell(F(\bar x;t),\bar y)}{\partial F_k} - \lambda(F_k(x;t) - F_k(x;0))\,,$$

*where $\bar\Theta^L$ is defined in (5).*

Note the term $-\lambda(F_k(x;t) - F_k(x;0))$, which constrains the network's output to stay close to its initialisation, is not present in the standard NTK dynamics (6).

**Evolution of the training objective.** As a side remark, we can see how the training objective $\mathcal{C}_s$ evolves during the training. First,

$$\partial_t \mathcal{L}_s(W(t)) = -\left\langle \nabla_F \hat{\ell}(F(X;t), Y) \cdot \partial_t F(X;t) \right\rangle_s$$

$$= \left\langle \bar{\Theta}(X, X') \nabla_F \hat{\ell}(F(X;t), Y) \cdot \nabla_F \hat{\ell}(F(X';t), Y') \right\rangle_{s\otimes s} - \lambda \left\langle \nabla_F \hat{\ell}(F(X;t), Y) \cdot \Delta F(X;t) \right\rangle_s,$$

where the notation $\langle g(Z, Z') \rangle_{s\otimes s}$ denotes $\frac{1}{m^2} \sum_{z \in s} \sum_{z' \in s} g(z, z')$. On the other hand, for the regularising term we have that

$$\partial_t \mathcal{R}(W(t)) = -\left\langle \nabla_F \hat{\ell}(F(X;t), Y) \cdot \Delta F(X;t) \right\rangle_s - 2\lambda \mathcal{R}(W(t)). \tag{9}$$

Thus, overall we get that

$$\partial_t \mathcal{C}_s(W(t))$$
$$= -\left\langle \bar{\Theta}(X, X') \nabla_F \hat{\ell}(F(X;t), Y) \cdot \nabla_F \hat{\ell}(F(X';t), Y') \right\rangle_{s\otimes s} - 2\lambda \left\langle \nabla_F \hat{\ell}(F(X;t), Y) \cdot \Delta F(X;t) \right\rangle_s - 2\lambda^2 \mathcal{R}(W(t)).$$

As a side remark, we note that if $\hat{\ell}$ is convex in $F$, then we always have that

$$\nabla_F \hat{\ell}(F(x;t), y) \cdot \Delta F(x;t) \geq \Delta \hat{\ell}(F(x;t), y) = \hat{\ell}(F(x;t), y) - \hat{\ell}(F(x;0), y).$$

In particular we get that

$$\partial_t \mathcal{R}(W(t)) \leq \langle \Delta \ell(F(x;t), Y) \rangle_s - 2\lambda \mathcal{R}(W(t)) = -\Delta \mathcal{C}_s(W(t)) - \lambda \mathcal{R}(W(t)).$$

Since $t \mapsto \Delta \mathcal{C}_s(W(t))$ is non-decreasing, we obtain that

$$\mathcal{R}(W(t)) \leq \frac{1}{\lambda} \Big( \mathcal{C}_s(W(0)) - \mathcal{C}_s(W(t)) \Big) (1 - e^{-\lambda t}),$$

from which it follows that $\mathcal{R}(W(t))$ can be controlled by the variation in $\mathcal{L}_s(W(t))$ as

$$\lambda \mathcal{R}(W(t)) \leq \frac{1 - e^{-\lambda t}}{2 - e^{-\lambda t}} \Big( \mathcal{L}_s(W(0)) - \mathcal{L}_s(W(t)) \Big).$$

### 4.2 General regulariser

We consider the case of a more general regularising term, which still leads to tractable training dynamics. Let $\rho : \mathbb{R}^+ \to \mathbb{R}^+$ be a differentiable strictly increasing function. Define

$$D(t) = \frac{1}{2} \|\Delta W(t)\|_F^2.$$

and let

$$\mathcal{R}(W(t)) = \rho(D(t)).$$

**Theorem 2.** *Consider a function $\rho : [0, \infty) \to [0, \infty)$ as above and assume that $\hat{\ell}$ is locally Lipschitz. Assume that the dynamics are given by*

$$\partial_t W(t) = -\nabla \mathcal{C}_s(W(t)) = -\nabla \mathcal{L}_s(W(t)) - \lambda \mathcal{R}(W(t)).$$

*Then, in the infinite-width limit (taken recursively layer by layer) we have*

$$\partial_t F_k(x;t) = -\left\langle \bar{\Theta}(x; X) \frac{\partial \hat{\ell}(F(X;t), Y)}{\partial F_k} \right\rangle_s - \lambda \rho'(D(t)) \Delta F_k(x;t);$$

$$\partial_t D(t) = -\sum_{k=1}^{q} \left\langle \Delta F_k(X;t) \frac{\partial \hat{\ell}(F(X;t), Y)}{\partial F_k} \right\rangle_s - 2\lambda \rho'(D(t)) D(t).$$

*Proof.* The proof is based on the following result, which we establish in Appendix A.1.

**Proposition 1.** *Fix a time horizon $T > 0$ and a depth $L$. Assume that for $t \in [0, T]$ and for all $l \in [1 : L]$*

$$\partial_t W_{ij}^l(t) = -\sum_{k=1}^{n_L} \left\langle \psi_{k;ij}^{L;l}(X;t) V_k(Z;t) \right\rangle_s - \lambda r(D(t);t) \Delta W_{ij}^l(t) \,,$$

*for some mappings $V : \mathcal{Z} \times \mathbb{R} \to \mathbb{R}^{n_L}$ and $r : [0, \infty)^2 \to \mathbb{R}$. Then we have that $U^L$ obeys the dynamics*

$$\partial_t U_k^L(x;t) = -\sum_{k'=1}^{n_L} \left\langle \Theta_{kk'}^L(x, X;t) V_{k'}(F(X;t), Y) \right\rangle_s - \lambda r(D(t);t) \Xi_k^L(x;t) \,.$$

*Moreover, if $D(t) = O(1)$ for all $t \in [0, T]$,*

$$\int_0^T \left\langle \|V(Z;t)\| \right\rangle_s \, \mathrm{d}t = O(1) \quad and \quad \int_0^T |r(D(t);t)| \mathrm{d}t = O(1) \,,$$

*if $\phi$ is $\gamma_\phi$-Lipschitz and $\beta_\phi$-smooth, then (in the infinite-width limit taken starting from the layer 1 and then going with growing index), for all $l \in [1 : L]$, $\Theta^l$ is constant during the training, and $\Xi^l = \Delta U^l$.*

To prove Theorem 2, we need to check that the assumptions of the above proposition hold when setting $r(D;t) = \rho'(D(t))$ and $V(z;t) = \nabla_F \ell(F(x;t), y)$. First, notice that $\mathcal{C}_s(t) \le \mathcal{C}_s(0) = \mathcal{L}_s(0)$ for all $t \ge 0$, as we are following the gradient flow. Now, with arbitrarily high probability (on the initialisation), we can find a finite upperbound $J$ for $\mathcal{L}_s(0)$, which holds when taking the infinite-width limit (as the network output becomes Gaussian). In particular, we have that since $J$ is independent of the width and we can write $J = O(1)$ (where the $O$ notation is referred to the infinite width limit). Now, we have that

$$\mathcal{R}(t) \le J/\lambda = O(1) \,.$$

Since $\rho$ is invertible, in particular for all $t \ge 0$ we have that

$$D(t) \le \rho^{-1}(J) = O(1) \,.$$

We prove in Lemma 1 (Appendix A.1) that $\Delta F(t) = O(1)$, and so we know that with arbitrarily high probability we can find a radius $J'$ such that $\|F(x;t)\| \le J'$ for all $t > 0$. In particular, the regularity of $\hat{\ell}$ implies that $V(z;t) = \nabla_F \hat{\ell}(F(x;t), y)$ is uniformly bounded for all $t > 0$. So, $\int_0^T \left\langle \|V(Z;t)\| \right\rangle_s \, \mathrm{d}t = O(1)$. Finally, we have that the $r(D;t)$ in the previous statement is $\rho'(D(t))$. Since $D(t)$ is bounded throughout the training and $\rho$ is locally Lipschitz we can bound the integral over $r$ and apply Proposition 1.

Finally, the dynamics for $D(t)$ follow from the chain rule. □

Clearly Theorem 1 is just a particular instance of Theorem 2, as we only need to set $\rho$ as the identity.

## 5 The example of least square regression

As a simple application of what we established, we study the evolution of a network under least square regression, namely when we have $\hat{\ell}(\hat{y}, y) = \frac{1}{2}(\hat{y} - y)^2$. The dynamics of the training are given by

$$\partial_t F(x;t) = -\left\langle \bar{\Theta}(x, X)(F(X;t) - Y) \right\rangle_s - \lambda(F(x;t) - F(x;0)) \,.$$

This is a linear ODE and can be solved exactly. For convenience we introduce the following notation. We let $\widetilde{\Theta}$ denote the NTK Gram matrix whose entries are $\bar{\Theta}(x, x')/m$, with $x$ and $x'$ ranging among the instances of the training sample $s$. Similarly $\widetilde{F}(t)$ and $\widetilde{Y}$ are the vectors made of the network's output and labels, for the datapoints in $s$. We thus have

$$\partial_t \widetilde{F}(t) = -\widetilde{\Theta}(\widetilde{F}(t) - \widetilde{Y}) - \lambda(\widetilde{F}(t) - \widetilde{F}(0)) \,,$$

which brings

$$\widetilde{F}(t) = \widetilde{F}(0) + (\mathrm{Id} - e^{-tV_\lambda})V_\lambda^{-1}\widetilde{\Theta}(\widetilde{Y} - \widetilde{F}(0))\,,$$

where $V_\lambda = \lambda\mathrm{Id} + \widetilde{\Theta}$, which is always invertible for $\lambda > 0$.

Note that asymptotically for large $t$ we have that

$$\widetilde{F}(t) \to \widetilde{F}(\infty) = \left(\mathrm{Id} - V_\lambda^{-1}\widetilde{\Theta}\right)\widetilde{F}(0) + V_\lambda^{-1}\widetilde{\Theta}\widetilde{Y}\,,$$

which is exactly what one would obtain optimising $\mathcal{C}_s$ in (1). Clearly, for small values of $\lambda$ the labels are almost perfectly approximated, as we have

$$\widetilde{F}(\infty) = \widetilde{Y} + \lambda\widetilde{\Theta}^{-1}(\mathrm{Id} + \lambda\widetilde{\Theta}^{-1})^{-1}(\widetilde{F}(0) - \widetilde{Y}) = \widetilde{Y} + O(\lambda)\,.$$

On the other hand, if $\lambda$ is very large then $\widetilde{F}(\infty) \simeq \widetilde{F}(0)$, as

$$\widetilde{F}(\infty) = \widetilde{F}(0) + \frac{\widetilde{\Theta}}{\lambda}\left(\mathrm{Id} + \frac{\widetilde{\Theta}}{\lambda}\right)^{-1}(Y - \widetilde{F}(0)) = \widetilde{F}(0) + O(1/\lambda)\,.$$

Once the evolution of $F$ on the training datapoints has been computed, one can directly evaluate the value of $F(x;t)$ for any input that is not in the training sample. Defining

$$\tau(x;t) = \left\langle \bar{\Theta}(x,X)(F(X;t) - Y)\right\rangle_s\,,\,^2$$

we have

$$F(x;t) = F(x;0) + (1 - e^{-\lambda t})\int_0^t \tau(x;t')e^{\lambda t'}\,\mathrm{d}t'\,.$$

## 6 An application to PAC-Bayesian training

A motivation to study dynamics in the form of (8) comes from the PAC-Bayesian literature. We consider a dataset $s$ made of i.i.d. draws from a distribution $\mu$ on $\mathcal{Z}$. We are seeking for parameters $W$ with a small population loss

$$\mathcal{L}_{\mathcal{Z}}(W) = \mathbb{E}_\mu[\ell(W)]\,.$$

As $\mu$ is unknown, we rely on the empirical loss $\mathcal{L}_s$ as a proxy for $\mathcal{L}_{\mathcal{Z}}$.

The PAC-Bayesian bounds (introduced in the seminal works of Shawe-Taylor & Williamson, 1997; McAllester, 1999; Seeger, 2002; Maurer, 2004; Catoni, 2004; 2007 – we refer to the recent surveys from Guedj, 2019; Alquier, 2021; Hellström et al., 2023 for a thorough introduction to PAC-Bayes) are generalisation guarantees that upperbound in high probability the population loss of stochastic architectures, in our case networks whose parameters $W$ are random variables (this means that every time that the network sees an input $x$, it draws $W$ from some distribution $Q$, and then evaluate $F(x)$ for this particular realisation of the parameters). The PAC-Bayesian bounds hold in expectation under the parameters law $Q$ (commonly referred to as *posterior*). Here is a concrete example of this kind of guarantees. Fix a probability measure $P$ and $\eta > 0$, for a bounded loss $\ell \in [0,1]$ we have (see, *e.g.*, Alquier, 2021)

$$\mathbb{E}_{W\sim Q}[\mathcal{L}_\mu(W)] \underset{1-\delta}{\leq} \mathbb{E}_{W\sim Q}[\mathcal{L}_s(W)] + \frac{1}{\sqrt{8m}}\left(\eta + \frac{\mathrm{KL}(Q|P) + \log(1/\delta)}{\eta}\right)\,, \tag{10}$$

where the inequality holds uniformly for every probability measure $Q$, in high probability (at least $1 - \delta$) with respect to the draw of $s$. The probability measure $P$ (typically called *prior*) in (10) is arbitrary, as long as it is chosen independently of the particular dataset $s$ used for the training.

Several studies (see Section 1 – this line of work started with Langford & Caruana, 2001, was reignited by Dziugaite & Roy, 2017 and then followed by a significant body of work by many authors) have proposed

---

[2]We remark that $\tau(x;t)$ can be computed: when averaging on $s$, $F(X;t)$ only takes as values the components of $\tilde{F}(t)$.

to train stochastic neural networks by optimising a PAC-Bayesian bound. A possible approach consists in considering the case when all the parameters of the network are independent Gaussian variables with unit variance (namely $W_{ij}^l \sim \mathcal{N}(\mathfrak{m}_{ij}^l, 1)$). The training then usually amounts to tune the means $\mathfrak{m}$. Typically, the initial values of the means are randomly initialised (we denote them as $\mathfrak{m}(0)$), and $P$ can be chosen as the distribution of the networks parameters at initialisation (Dziugaite & Roy, 2017), namely a multivariate normal with the identity as covariance matrix and $\mathfrak{m}(0)$ as mean vector. In this way, one gets that

$$\mathrm{KL}(Q|P) = \frac{1}{2}\|\mathfrak{m}(t) - \mathfrak{m}(0)\|_{\mathrm{F}}^2\,.$$

Defining $\bar{\mathcal{L}}_s(\mathfrak{m}) = \mathbb{E}_{W \sim \mathcal{N}(\mathfrak{m}, \mathrm{Id})}[\mathcal{L}_s(W)]$, we see that using the bound (10) as training objective is equivalent to optimise

$$\mathcal{C}_s(\mathfrak{m}) = \bar{\mathcal{L}}_s(\mathfrak{m}) + \frac{1}{2\eta\sqrt{8m}}\|\mathfrak{m} - \mathfrak{m}(0)\|_{\mathrm{F}}^2\,, \tag{11}$$

which is exactly in the form of (7) with $\lambda = 1/(\eta\sqrt{8m})$. More generally, many PAC-Bayesian bounds are not linear in the KL therm. However, they can still fit in our framework (with a general reguliriser $\rho$ as in Section 4.2) as long as they are in the form

$$\mathbb{E}_{W \sim Q}[\mathcal{L}_\mu(W)] \underset{1-\delta}{\leq} \mathbb{E}_{W \sim Q}[\mathcal{L}_s(W)] + \tilde{\rho}(\mathrm{KL}(Q|P))$$

for some strictly increasing and differentiable $\tilde{\rho}$. This is for instance the case for the training objective used for the PAC-Bayesian training by Dziugaite & Roy (2018).

While significant experimental work has focused on PAC-Bayesian training methods and achieved promising results, to our knowledge the literature lacks of rigorous theoretical studies of these training dynamics. Since the NTK formulation has already been successfully used for the study of gradient descent in the unregularised case, we anticipate that the closed-form expression for the network's evolution that we derived could help study various properties (such as rates of convergence, convergence to global/local minima, etc.).

**Training of wide and shallow stochastic networks**

Clerico et al. (2023b) has recently shown that, when considering the infinite-width limit for a single-hidden-layer stochastic network, a close form for $\hat{\mathcal{L}}_s(\mathfrak{m})$ can be computed explicitly, and one can actually see the stochastic network as a deterministic one (with a different activation function), where the means $\mathfrak{m}$ are the trainable parameters. We summarise and rephrase the results of Clerico et al. (2023b), and show explicitly how exact continuous dynamics can be established via our results, in the infinite-width limit.

We focus on a binary classification problem (i.e., $\mathcal{Y} = \{\pm 1\}$), where we assume that all the inputs $x$ are normalised so that $\|x\| = \sqrt{n_0}$ (i.e., $\mathcal{X} = S^{n_0-1}(\sqrt{n_0})$, the sphere of radius $\sqrt{n_0}$ in $\mathbb{R}^{n_0}$). We consider a stochastic network with a single hidden layer and one-dimensional output, and Lipschitz and smooth activation $\phi$,

$$F(x) = \frac{1}{\sqrt{n}}\sum_{j=1}^n W_j^2 \phi(U_j^1(x))\,; \qquad U_i^1(x) = \frac{1}{\sqrt{n_0}}\sum_{j=1}^{n_0} W_{ij}^1 x_j\,.$$

The prediction of the network is the sign of the output. The stochastic parameters can be rewritten as

$$W^l = \zeta^l + \mathfrak{m}^l\,,$$

where $\zeta^l$ is a matrix of the same dimension of $W^l$, whose components are all independent standard normals (resampled every time that a new input is fed to the network) and $\mathfrak{m}^l$ is a matrix of trainable parameters (the means of $W^l$). We assume that the components of $\mathfrak{m}^l$ are all randomly initialised as independent draws from a standard normal distribution. In this setting, Clerico et al. (2023b) showed that in the infinite-width limit $n \to \infty$

$$F(x) \sim \mathcal{N}(M^2(x), Q^2(x))\,,^{[3]} \tag{12}$$

---

[3]The limit is in distribution with respect to the intrinsic stochasticity of the $\zeta$'s, and in probability with respect to the random initialisation.

where $M^2$ and $Q^2$ are the output mean and variance in the limit.

When the training objective is in the form (11), the network is constraint to remain close to its initialisation. Interestingly, in this lazy training regime $Q^2$ stays constant to its initial value, which is deterministic (with respect to the initialisation randomness) and independent of $x$ when $\mathcal{X}$ is a sphere. We can thus define $\sigma > 0$ such that $Q^2(x; t) = \sigma^2$, for all $x \in \mathcal{X}$ and $t$. We refer to the Appendix A.2 for details.

Following the derivation of Clerico et al. (2023b) we note that $M^2$ can actually be seen as the output of a neural network with parameters $\mathfrak{m}$, whose activation function is $\psi$, defined as a Gaussian convolution of $\phi$,

$$\psi(u) = \mathbb{E}_{\zeta \sim \mathcal{N}(0,1)}[\phi(\zeta + u)] \,.$$

Concretely, this means that

$$M^2(x) = \frac{1}{\sqrt{n}} \mathfrak{m}^2 \psi(M^1(x)) \,; \qquad M^1(x) = \frac{1}{\sqrt{n_0}} \mathfrak{m}^1 x \,.$$

Now, for a loss function $\hat{\ell}(F, z)$, we can define the expected loss

$$\bar{\ell}(M, z) = \mathbb{E}_{\zeta \sim \mathcal{N}(0,1)}[\hat{\ell}(\sigma\zeta + M, z)] \,.$$

In this way, we have that the expected empirical loss $\bar{\mathcal{L}}_s(\mathfrak{m})$ appearing in (11) is given by

$$\bar{\mathcal{L}}_s(\mathfrak{m}) = \frac{1}{m} \sum_{z \in s} \bar{\ell}(M^2(x), y) \,.$$

Hence, optimising the PAC-Bayes bound (10) induces the dynamics

$$\partial_t M^2(x; t) = -\left\langle \bar{\Theta}(x; X) \frac{\partial \bar{\ell}(M^2(X; t), Y)}{\partial M^2} \right\rangle_s - \frac{1}{\eta\sqrt{8m}} \Delta M^2(x; t) \,, \tag{13}$$

where $\bar{\Theta}(x, x') = \nabla_{\mathfrak{m}} M^2(x; 0) \cdot \nabla_{\mathfrak{m}} M^2(x'; 0)$ is given by

$$\bar{\Theta}(x, x') = \Sigma(x, x') + \frac{x \cdot x'}{n_0} \mathbb{E}[\dot{\psi}(\zeta)\dot{\psi}(\zeta')] \,; \qquad \Sigma(x, x') = \mathbb{E}[\psi(\zeta)\psi(\zeta')] \,,$$

with $(\zeta, \zeta') \sim \mathcal{N}\left(0, \frac{1}{n_0}\begin{pmatrix} n_0 & x \cdot x' \\ x \cdot x' & n_0 \end{pmatrix}\right)$ and $\dot{\psi}$ denoting the derivative of $\psi$.

**Misclassification loss.** A common choice is to set $\hat{\ell}(F, z) = 1$ if sign $F(x) \neq y$, and 0 otherwise, that is the so-called misclassification loss. In such a case we can easily derive that

$$\bar{\ell}(M, z) = \mathbb{P}_{\zeta \sim \mathcal{N}(0,1)}\left(\zeta > \frac{yM(x)}{\sigma}\right) = \frac{1}{2}\left(1 - \operatorname{erf}\left(\frac{yM(x)}{\sigma\sqrt{2}}\right)\right) \,.$$

It follows that (13) now reads

$$\partial_t M^2(x; t) = \left\langle Y\bar{\Theta}(x; X) \frac{e^{-M^2(X; t)/(2\sigma^2)}}{\sigma\sqrt{2\pi}} \right\rangle_s - \frac{1}{\eta\sqrt{8m}} \Delta M^2(x; t) \,.$$

This does not have a simple close-form solution, and can only be solved using numerical integrators.

**Quadratic loss.** In order to obtain simpler dynamics we consider the loss $\hat{\ell}(F, z) = (1 - yF(x))^2$. Note that this quadratic loss is unbounded, and so a generalisation bound such as (10) is not guaranteed to hold. However, the quadratic loss is always greater than the misclassification loss, so the RHS of (10) is still a valid generalisation upperbound for the misclassification population loss. This choice of $\hat{\ell}$ yields the dynamics

$$\partial_t M^2(x; t) = -2\left\langle \bar{\Theta}(x; X)(M^2(x; t) - Y)\right\rangle_s - \frac{1}{\eta\sqrt{8m}} \Delta M^2(x; t) \,.$$

Proceeding as in Section 5, and again using a tilde to denote vectors and matrices indexed on the training sample $s$, we get that

$$\widetilde{M}^2(t) = \widetilde{M}^2(0) + \left(\mathrm{Id} - e^{-2tV_{\lambda/2}}\right) V_{\lambda/2}^{-1}\widetilde{\Theta}(\widetilde{Y} - \widetilde{M}^2(0)),$$

where $V_{\lambda/2} = \lambda\mathrm{Id}/2 + \widetilde{\Theta}$ and $\lambda = 1/(\eta\sqrt{8m})$. Asymptotically, for large $t$, $\widetilde{M}^2$ will approach

$$\widetilde{M}^2(\infty) = \widetilde{M}^2(0) + V_{\lambda/2}^{-1}\widetilde{\Theta}(\widetilde{Y} - \widetilde{M}^2(0)).$$

For large $t$, the empirical loss approaches

$$\bar{\mathcal{L}}_\infty = \frac{1}{m}\|\widetilde{M}^2(0) - \widetilde{Y}\|^2_{(\lambda/2)^2 V_{\lambda/2}^{-2}},$$

where for a positive definite matrix $A$ we define $\|v\|_A^2 = v^\top A v$. We note that the eigenvalues of $(\lambda/2)^2 V_{\lambda/2}^{-2}$ are in the form

$$\alpha_i = \frac{\lambda^2/4}{(\lambda/2 + \theta_i)^2},$$

where the $\theta_i$'s are the eigenvalues of $\widetilde{\Theta}$. In practice, we can expect the largest eigenvalues of $\widetilde{\Theta}$ to be of order 1 (*i.e.*, $\max_i \theta_i \sim 1$) when the sample size $m$ grows to infinity (Murray et al., 2023). Since $\lambda \sim 1/\sqrt{m}$, we get that the network will be able to reach a small empirical loss (of order $\lambda^2 \sim 1/m$) if $\widetilde{M}^2(0) - \widetilde{Y}$ lies in eigenspaces of $\widetilde{\Theta}$ where the eigenvalues $\theta_i \sim 1 \gg 1/\sqrt{m}$.

On the other hand, for large $t$ the regularising term $\mathcal{R}$ will approach $\mathcal{R}_\infty$, which from (9) must satisfy

$$\frac{2}{m}\Delta\widetilde{M}(\infty) \cdot (\widetilde{M}(\infty) - \widetilde{Y}) = -2\lambda\mathcal{R}_\infty.$$

From this we can derive that

$$\mathcal{R}_\infty = \frac{1}{2m}\|\widetilde{M}^2(0) - \widetilde{Y}\|^2_{V_{\lambda/2}^{-2}\widetilde{\Theta}}.$$

Here, the eigenvalues of $V_{\lambda/2}^{-2}\widetilde{\Theta}$ are of the form

$$\beta_i = \frac{\theta_i}{(\lambda/2 + \theta_i)^2}.$$

If we are in the regime where the datapoints are such that $\widetilde{M}^2(0) - \widetilde{Y}$ again lies where $\theta_i \sim 1$, then we can expect $\mathcal{R}_\infty \sim 1$. This means that if we are able to learn well the labels while optimising the PAC-Bayesian bound, we are ensured that the KL term is of order 1, and so the objective (11) will be of order $1/\sqrt{m}$, resulting in a non-vacuous bound. We argue that this is what happens when data comes from a *reasonable* underlying distribution, matching the implicit regularisation induced by the NTK.

We finally note that the $\beta_i$'s are small also when $\theta_i \ll 1/\sqrt{m}$. However this is due to the fact that these directions are not promoted by the NTK dynamics and so if $\widetilde{M}^2(0) - \widetilde{Y}$ completely lies in eigenspaces with very small eigenvalues of $\widetilde{\Theta}$, then the network will essentially stay fixed to its initial configuration, keeping a small penalty, but also not improving its performance.

## 7 Conclusion

We established explicit dynamics for infinitely wide fully connected networks trained to optimise a regularised objective, where the regularisation pushes the parameters to stay close to their initialisation. Under this regime we show that the model undergoes linearised dynamics during the training, which turns out to be a regularised version of the standard NTK evolution.

Our analysis follows similar ideas to the NTK convergence proof of Jacot et al. (2018) and presents the first regularised NTK analysis that can also be applied to PAC-Bayesian training. We conjecture that stronger

follow-up results could be derived, for instance, following the approach of Lee et al. (2019) to show that the convergence holds when the infinite width limit is taken for all the hidden layers simultaneously, and to study discretised dynamics.

We also anticipate that further analytical and empirical studies of the induced PAC-Bayesian dynamics might be of interest to shed some light on generalisation-driven training of neural networks.

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

# A  Appendix

## A.1  Proof of Proposition 1

**Lemma 1.** *Assume that, when the infinite-width limit is taken recursively layer by layer, $D(t) = O(1)$. Assume that $\phi$ is $\gamma_\phi$-Lipschitz, then we have that for each layer $l$*

$$\|\Delta U^l(x; t)\| = O(1)\,.$$

*Proof.* We have that

$$\|\Delta U^1(x; t)\| \leq \frac{1}{\sqrt{n_0}} \|\Delta W^0(t)\| \|x\| = O(1)\,.$$

Then, recall that at initialisation the components of $U^l(x, 0)$ are independent normal distributions, with mean 0 and variance $\Sigma^l(x, x)$, as defined in (4). So, we have that all the $\phi(U_i^l(x; 0))^2$'s are independent and equally distributed, with finite variance (thanks to the Lipschitzness of $\phi$). By the standard CLT we get that

$$\frac{1}{n_l} \|\phi(U^l(x; 0))\|^2 = \frac{1}{n_l} \sum_{i=1}^{n_l} \phi(U_i^l(x; 0))^2 \sim \frac{1}{n_l} \sum_{i=1}^{n_l} \mathbb{E}[\phi(U_i^l(x; 0))^2] = O(1)\,,$$

and so $\|\phi(U^l(x; 0))\| = O(\sqrt{n_l})$. Now, using the Lipschitzness of $\phi$ we get

$$\|\Delta U^{l+1}(x; t)\| \leq \frac{1}{\sqrt{n_l}} \left( \|W^{l+1}(0)\| \gamma_\phi \|\Delta U^l(x; t)\| + \|\Delta W^{l+1}(t)\| \|\phi(U^l(x; 0))\| + \gamma_\phi \|\Delta U^l(x; t)\| \|\Delta W^{l+1}(t)\| \right)$$

$$= O\left( 1 + \left( 1 + \sqrt{\frac{n_{l+1}}{n_l}} \right) \|\Delta U^l(x; t)\| \right)\,,$$

where we used that $\|W^{l+1}(0)\| = O(\sqrt{n_l} + \sqrt{n_{l+1}})$, a classical result in random matrix theory (Vershynin, 2012). Assuming that the limit is taken layer after layer, we have that $n_{l+1}/n_l \to 0$ and so

$$\|\Delta U^{l+1}(x; t)\| = O(1)$$

by induction. □

For the rest of this section, we define $D^l(t)$ as

$$D^l(t) = \frac{1}{2} \sum_{l'=1}^{l} \|W^{l'}(t) - W^{l'}(0)\|_{\mathrm{F}}^2\,.$$

Now, we restate and prove Proposition 1.

**Proposition 1.** *Fix a time horizon $T > 0$ and a depth $L$. Assume that for $t \in [0, T]$ and for all $l \in [1 : L]$*

$$\partial_t W_{ij}^l(t) = -\sum_{k=1}^{n_L} \left\langle \psi_{k;ij}^{L;l}(X; t) V_k(Z; t) \right\rangle_s - \lambda r(D^L(t); t) \Delta W_{ij}^l(t)\,,$$

*for any mappings $V : \mathcal{Z} \times \mathbb{R} \to \mathbb{R}^{n_L}$ and $r : [0, \infty)^2 \to \mathbb{R}$. Then we have that $U^L$ obeys the dynamics*

$$\partial_t U_k^L(x; t) = -\sum_{k'=1}^{n_L} \left\langle \Theta_{kk'}^L(x, X; t) V_{k'}(F(X; t), Y) \right\rangle_s - \lambda r(D^L(t); t) \Xi_k^L(x; t)\,.$$

*Moreover, if $D^L(t) = O(1)$ for all $t \in [0, T]$,*

$$\int_0^T \langle \|V(Z; t)\| \rangle_s \, \mathrm{d}t = O(1) \quad \text{and} \quad \int_0^T |r(D^L(t); t)| \mathrm{d}t = O(1)\,,$$

*if $\phi$ is $\gamma_\phi$-Lipschitz and $\beta_\phi$-smooth, then (in the infinite-width limit taken starting from the layer 1 and then going with growing index), for all $l \in [1 : L]$, $\Theta^l$ is constant during the training, and $\Xi^l = \Delta U^l$.*

*Proof.* The first statement follows directly from the chain rule. For the second statement, we proceed by induction, taking inspiration in the original NTK proof of Jacot et al. (2018). For $L = 1$ the model is linear and the statement holds. Now, assume that the statement holds for networks of depth $L$, we want to show that it is true also for architectures of depth $L + 1$. We hence consider a network of depth $L + 1$ following the dynamics

$$\partial_t W_{ij}^l(t) = -\sum_{k=1}^{n_{L+1}} \left\langle \psi_{k;ij}^{L+1;l}(X;t) V_k(Z;t) \right\rangle_s - \lambda r(D^{L+1}(t);t)\Delta W_{ij}^l(t)$$

and satisfying $\int_0^T \langle \|V(Z;t)\| \rangle_s \, \mathrm{d}t = O(1)$, $\int_0^T r(D^{L+1}(t);t)\mathrm{d}t = O(1)$, $D^{L+1}(t) = O(1)$, for all $t \in [0, T]$.

We note that for $l \in [1 : L]$

$$\psi_{k;ij}^{L+1;l}(x;t) = \frac{\partial U_k^{L+1}(x;t)}{\partial W_{ij}^l} = \sum_{k'=1}^{n_L} \frac{\partial U_k^{L+1}(x;t)}{\partial U_{k'}^L} \psi_{k';ij}^{L;l}(x;t) = \frac{1}{\sqrt{n_L}} \sum_{k'=1}^{n_L} W_{kk'}^{L+1}(t)\dot{\phi}(U_{k'}^L(x;t))\psi_{k';ij}^{L;l}(x;t) \, .$$

We define

$$\widetilde{V}_{k'}(z;t) = \frac{1}{\sqrt{n_L}} \sum_{k=1}^{n_{L+1}} W_{kk'}^{L+1}(t)\dot{\phi}(U_{k'}^L(x;t))V_k(z;t)$$

and

$$\widetilde{r}(D;t) = r(\|\Delta W^{L+1}(t)\|_{\mathrm{F}}^2/2 + D;t) \, ,$$

so that we can rewrite the dynamics for $l \in [1 : L]$ as

$$\partial_t W_{ij}^l(t) = -\sum_{k'=1}^{n_L} \left\langle \psi_{k';ij}^{L;l}(X;t)\widetilde{V}_{k'}(Z;t) \right\rangle_s - \lambda\widetilde{r}_t(D^L(t))\Delta W_{ij}^l(t) \, .$$

We have that

$$\int_0^T \left\langle \|\widetilde{V}(Z;t)\| \right\rangle_s \, \mathrm{d}t \le \frac{\gamma_\phi}{\sqrt{n_L}} \int_0^T \|W^{L+1}(t)\| \langle \|V(Z;t)\| \rangle_s \, \mathrm{d}t \, ,$$

which is of order $O(1)$ if $\|W^{L+1}(t)\|/\sqrt{n_L}$ is. This is indeed the case, as we know that $\|\Delta W^{L+1}(t)\| \le \|\Delta W^{L+1}(t)\|_{\mathrm{F}} = O(1)$, and $\|W^{L+1}(0)\| = O(\sqrt{n_{L+1}} + \sqrt{n_L})$ (this is a classical result on random matrix theory; see for instance Vershynin, 2012). Moreover, for each $t$ we have that $\widetilde{r}(D^L(t);t) = r(D^{L+1}(t);t)$, so that in particular

$$\int_0^T |\widetilde{r}(D^L(t);t)|\mathrm{d}t = \int_0^T |r(D^{L+1}(t);t)|\mathrm{d}t = O(1) \, .$$

We can hence apply the inductive hypothesis to the sub-network made of the first $L$ layers, and obtain that, for $l \in [1 : L]$, $\Theta^l$ stays constant during the training and $\Xi^l = \Delta U^l$. We also recall that at initialisation the kernels $\Theta^l$ are diagonal, and so we have that for all $t \in [0 : T]$

$$\Theta_{kk'}^l(x, x';t) = \delta_{kk'}\bar{\Theta}^l(x, x') \, .$$

Now, in order to conclude we need to check that the claim holds also for the last layer. We have

$$\begin{aligned}
\Theta_{kk'}^{L+1}(x, x';t) &= \sum_{l=1}^{L+1}\sum_{k=1}^{n_l} \psi_k^{L+1;l}(x;t) \cdot \psi_{k'}^{L+1;l}(x';t) \\
&= \psi_k^{L+1;L+1}(x;t) \cdot \psi_{k'}^{L+1;L+1}(x';t) + \sum_{j,j'=1}^{n_L} \frac{\partial U_k^{L+1}(x;t)}{\partial U_j^L}\frac{\partial U_{k'}^{L+1}(x';t)}{\partial U_{j'}^L}\Theta_{jj'}^L(x, x';t) \\
&= \psi_k^{L+1;L+1}(x;t) \cdot \psi_{k'}^{L+1;L+1}(x';t) + \sum_{j=1}^{n_L} \frac{\partial U_k^{L+1}(x;t)}{\partial U_j^L}\frac{\partial U_{k'}^{L+1}(x';t)}{\partial U_j^L}\bar{\Theta}^L(x, x') \, .
\end{aligned}$$

Let us denote as $u_k(x;t)$ the vector with components $u_{k;j}(x;t) = \frac{\partial U_k^{L+1}(x;t)}{\partial U_j^L}$. We easily see that

$$\|u_k(x;t)\| \leq \frac{\gamma_\phi}{\sqrt{n_L}} \|W_{k\cdot}^{L+1}(t)\| \leq \frac{\gamma_\phi}{\sqrt{n_L}} \|W_{k\cdot}^{L+1}(0)\| + \frac{\gamma_\phi}{\sqrt{n_L}} \|\Delta W^{L+1}(t)\| = O(1) \,,$$

where we used that $\|W_{k\cdot}^{L+1}(0)\| = \sqrt{n_L}$ and that the norm of the row of a matrix is always bounded by the Frobenius norm of the matrix. On the other hand, we have that

$$\|\Delta u_k(x;t)\| \leq \frac{\gamma_\phi}{\sqrt{n_L}} \|\Delta W^{L+1}(t)\| + \frac{\beta_\phi}{\sqrt{n_L}} \|W_{k\cdot}^{L+1}(0)\|_\infty \|\Delta U^L(t)\| \,,$$

where we used that $\phi$ is $\gamma_\phi$-Lipschitz and $\beta_\phi$-smooth. We know that $\|\Delta W^{L+1}(t)\| = O(1)$ as $D^{L+1} = O(1)$. Moreover $\|W_{k\cdot}^{L+1}(0)\|_\infty$ behaves as the maximum of $n_L$ independent standard normals, that is $\|W_{k\cdot}^{L+1}(0)\|_\infty = O(\sqrt{\log n_L})$ (Boucheron et al., 2013). On the other hand, $\|\Delta U^L(t)\| = O(1)$ by Lemma 1. We hence easily conclude that

$$\Delta \left( \sum_{j=1}^{n_L} \frac{\partial U_k^{L+1}(x;t)}{\partial U_j^L} \frac{\partial U_{k'}^{L+1}(x';t)}{\partial U_j^L} \bar{\Theta}^L(x,x') \right) = O(\sqrt{\log(n_L)/n_L}) \,.$$

Now to show that the variation of the NTK vanishes during the training we only need to control $\Delta(\psi_k^{L+1;L+1}(x;t) \cdot \psi_{k'}^{L+1;L+1}(x';t))$. First, notice that

$$\|\psi_k^{L+1;L+1}(x;0)\| \leq \sqrt{\bar{\Theta}^{L+1}(x,x')} = O(1) \,.$$

Moreover, we have that $\psi_k^{L+1;L+1}(x;t) = \frac{\delta_{ik}}{\sqrt{n_L}} \phi(U_j^L(x;t))$ and so

$$\|\Delta \psi_k^{L+1;L+1}(x;t)\| \leq \frac{\gamma_\phi}{\sqrt{n_L}} \|\Delta U^L(x;t)\| = O(1/\sqrt{n_L}) \,.$$

We thus deduce that

$$\Delta(\psi_k^{L+1;L+1}(x;t) \cdot \psi_{k'}^{L+1;L+1}(x';t)) = O(1/\sqrt{n_L})$$

and so

$$\Delta \Theta_{kk'}^{L+1}(x,x';t) = O(\sqrt{\log(n_L)/n_L}) \,,$$

which shows that in the infinite-width limit the NTK stays constant during the training.

Now we are left with showing that $\Xi^{L+1} = \Delta U^{L+1}$. Recalling the notation $u_{k;k'}(x;t) = \frac{\partial U_k^{L+1}(x;t)}{\partial U_{k'}^L}$, we can write

$$\Xi_k^{L+1}(x;t) = \sum_{l=1}^{L+1} \psi_k^{L+1;l} \cdot \Delta W^l(t) = \psi_k^{L+1;L+1}(x;t) \cdot \Delta W^{L+1}(t) + u_k(x;t) \cdot \Xi^L(x;t) \,.$$

Using the induction hypothesis $\Xi^L = \Delta U^L$, we get that

$$\Xi_k^{L+1}(x;t) = \psi_k^{L+1;L+1}(x;t) \cdot \Delta W^{L+1}(t) + u_k(x;t) \cdot \Delta U^L(x;t) \,.$$

Using that $\partial_t U_k^{L+1} = \psi_k^{L+1;L+1} \cdot \partial_t W^{L+1} + u_k \cdot \partial_t U^L$, we can write

$$\Xi_k^{L+1}(x;t) - \Delta U_k^{L+1}(x;t)$$
$$= \int_0^t \left( \psi_k^{L+1;L+1}(x;t) - \psi_k^{L+1;L+1}(x;t') \right) \cdot \partial_t W^{L+1}(t') \mathrm{d}t' + \int_0^t \left( u_k(x;t) - u_k(x;t') \right) \cdot \partial_t U^L(x;t') \mathrm{d}t' \,.$$

In particular,

$$|\Xi_k^{L+1}(x;t) - \Delta U_k^{L+1}(x;t)|$$
$$\leq 2 \sup_{t' \in [0,t]} \|\Delta \psi_k^{L+1;L+1}(x;t')\| \int_0^t \|\partial_t W^{L+1}(t')\|_F \mathrm{d}t' + 2 \sup_{t' \in [0,t]} \|\Delta u_k(x;t')\| \int_0^t \|\partial_t U^L(x;t')\| \mathrm{d}t' \,.$$

From what we have shown already, we know that

$$\sup_{t' \in [0,t]} \|\Delta \psi_k^{L+1;L+1}(x;t')\| = O(1/\sqrt{n_L}) \qquad \text{and} \qquad \sup_{t' \in [0,t]} \|\Delta u_k(x;t')\| = O(\sqrt{\log(n_L)/n_L}) \,,$$

so we are left with checking that the last two integrals are of order $O(1)$ in order to conclude. First, we have

$$\partial_t W_{ij}^{L+1}(t) = -\frac{1}{\sqrt{n_L}} \left\langle \phi(U_j^L(X;t)) V_i(X;t) \right\rangle_s - \lambda r(D^{L+1}(t);t) \Delta W_{ij}^{L+1}(t) \,.$$

Since $\|\Delta \phi(U^L(x;t))\| = O(1)$, we have $\|\phi(U^L(x;t))\| = O(\sqrt{n_L})$, and we can define

$$K' = \frac{1}{\sqrt{n_L}} \sup_{x' \in s} \|\phi(U^L(x';t))\| = O(1) \,.$$

We have thus obtained

$$\|\partial_t W^{L+1}(t)\| \leq K' \left\langle \|V(Z;t)\| \right\rangle_s + \lambda |r(D^{L+1}(t);t)| \|\Delta W^{L+1}(t)\| \,.$$

So,

$$\int_0^t \|\partial_t W^{L+1}(t')\| \mathrm{d}t' \leq K' \int_0^t \left\langle \|V(Z;t')\| \right\rangle_s \mathrm{d}t' + \sup_{t' \in [0,t]} \|\Delta W^{L+1}(t')\| \lambda \int_0^t |r(D^{L+1}(t');t')| \mathrm{d}t' = O(1) \,,$$

since both integrals in the RHS can be controlled by hypothesis.

Now we just need to control $\int_0^t \|\partial_t U^L(x;t')\| \mathrm{d}t'$. Defining $K(x) = \sup_{x' \in s} |\bar{\Theta}^L(x,x')|$, we easily get that

$$\|\partial_t U^L(x;t)\| \leq K(x) \left\langle \|\widetilde{V}(Z;t)\| \right\rangle_s + \lambda |\widetilde{r}(D^L(t);t)| \|\Delta U^L(x;t)\| \,.$$

We have established that $\int_0^t \left\langle \|\widetilde{V}(Z;t')\| \right\rangle_s \mathrm{d}t' = O(1)$ and $\sup_{t' \in [0,t]} \|\Delta U^L(x;t')\| = O(1)$. In particular, since by assumption $\int_0^t |r(D^{L+1}(t');t')| \mathrm{d}t' = O(1)$, we have that indeed

$$\int_0^t \|\partial_t U^L(x;t')\| \mathrm{d}t' = O(1) \,.$$

With this last step we have shown that in the infinite width limit

$$\Xi^{L+1}(x;t) = \Delta U^{L+1}(x;t) \,,$$

which concludes the proof. □

## A.2 Variance of the wide stochastic network

From Clerico et al. (2023b) (eq. 5 therein, applied to a one-dimensional output) we know that the output's variance $Q^2$ of a shallow wide stochastic network is given by

$$Q^2(x;t) = \frac{1}{n} \sum_{j=1}^n (1 + (\mathfrak{m}_j^2(t))^2) \xi(M_j^1(x;t)) - \frac{1}{n} \sum_{j=1}^n (\mathfrak{m}_j^2(t))^2 \psi(M_j^1(x;t))^2 \,,$$

where we define $\xi(u) = \mathbb{E}_{\zeta \sim \mathcal{N}(0,1)}[\phi(\zeta + u)^2]$ and we recall that $\psi(u) = \mathbb{E}_{\zeta \sim \mathcal{N}(0,1)}[\phi(\zeta + u)]$.

We now consider an initialisation where each component of $\mathfrak{m}^1$ and $\mathfrak{m}^2$ is sampled independently from $\mathcal{N}(0,1)$. Then, since for any input $x$ we have $\|x\| = \sqrt{n_0}$, each component of $M_j^1(0)$ is distributed (with respect to the initialisation's randomness) as a standard normal distribution. Since $\mathfrak{m}^2$ is independent of $M^1(0)$ we can easily derive that, in the limit $n \to \infty$,

$$Q^2(x;0) = 2\mathbb{E}_{\zeta \sim \mathcal{N}(0,1)}[\xi(\zeta)] - \mathbb{E}_{\zeta \sim \mathcal{N}(0,1)}[\psi(\zeta)^2] = \sigma^2 \,,$$

which is a deterministic value.

We now show here that $Q^2$ keeps constant during the training. We have that

$$\|\Delta M^1(x;t)\| \leq \frac{1}{\sqrt{n_0}}\|\Delta\mathfrak{m}^1(t)\|\|x\| = \|\Delta\mathfrak{m}^1(t)\| = O(1)\,.$$

Now, let $u_j(t) = 1 + \mathfrak{m}_j^2(t)^2$. We have that

$$\Delta\left(\sum_{j=1}^{n}(1 + (\mathfrak{m}_j^2(t))^2)\xi(M_j^1(x;t))\right) = u(0)\cdot\Delta\xi(M^1(x;t)) + \Delta u(t)\cdot\xi(M^1(x;0)) + \Delta u(t)\cdot\Delta\xi(M^1(x;t))\,.$$

Let us start by the first term. We have that

$$\Delta\xi(M_j^1(x;t)) = \mathbb{E}_{\zeta\sim\mathcal{N}(0,1)}\left[(2\phi(M_j^1(x;0) + \zeta) + \Delta\phi(M_j^1(x;t) + \zeta))\Delta\phi(M_j^1(x;t) + \zeta)\right]\,,$$

and so (recalling that we are assuming that $\phi$ is $C_\phi$ Lipschitz)

$$|u(0)\cdot\Delta\xi(M^1(x;t))|$$

$$\leq 2\left|\sum_{j=1}^{n}\mathbb{E}_{\zeta\sim\mathcal{N}(0,1)}\left[u_j(0)\phi(M_j^1(x;0) + \zeta)\Delta\phi(M_j^1(x;t) + \zeta)\right]\right| + \left|\sum_{j=1}^{n}\mathbb{E}\left[u_j(0)\Delta\phi(M_j^1(x;t) + \zeta)^2\right]\right|$$

$$\leq 2C_\phi\|\Delta M^1(x;t)\|\mathbb{E}_{\zeta\sim\mathcal{N}(0,1)}\left[\sum_{j=1}^{n}\|u_j(0)\phi(M_j^1(x;0) + \zeta)\|^2\right]^{1/2} + C_\phi^2\|\Delta M^1(x;t)\|^2\|u(0)\|\,.$$

For large $n$ we have that

$$\frac{1}{n}\sum_{j=1}^{n}\|u_j(0)\phi(M_j^1(x;0) + \zeta)\|^2 \to 2\mathbb{E}_{\zeta'\sim\mathcal{N}(0,1)}[\phi(\zeta' + \zeta)^2] = O(1)$$

(in probability with respect to the random initialisation) and $\|u(0)\| \to \sqrt{2n}$. Since $\|\Delta M^1(x;t)\| = O(1)$, we have that

$$u(0)\cdot\Delta\xi(M^1(x;t)) = O(\sqrt{n})\,.$$

Proceeding similarly we get that

$$|\Delta u(t)\cdot\xi(M^1(x;0))| \leq 2\left(\sum_{j=1}^{n}\mathfrak{m}_j^2(0)^2\xi(M^1(x;0))\right)^{1/2}\|\Delta\mathfrak{m}^2(t)\| + \|\Delta\mathfrak{m}^2(t)\|_4^2\|\xi(M^1(x;0))\|$$

$$\leq 2\left(\sum_{j=1}^{n}\mathfrak{m}_j^2(0)^2\xi(M^1(x;0))\right)^{1/2}\|\Delta\mathfrak{m}^2(t)\| + \|\Delta\mathfrak{m}^2(t)\|_2^2\|\xi(M^1(x;0))\| = O(\sqrt{n})\,.$$

Finally, we have that

$$\Delta u(t)\cdot\Delta\xi(M^1(x;t)) = 2\sum_{j=1}^{n}\mathfrak{m}_j^2(0)\Delta\mathfrak{m}_j^2(t)\Delta\xi(M_j^1(t)) + \sum_{j=1}^{n}(\Delta\mathfrak{m}_j^2(t))^2\Delta\xi(M_j^1(t))\,.$$

We have that

$$\left|\sum_{j=1}^{n}\mathfrak{m}_j^2(0)\Delta\mathfrak{m}_j^2(t)\Delta\xi(M_j^1(t))\right|$$

$$\leq 2C_\phi\mathbb{E}_{\zeta\sim\mathcal{N}(0,1)}\left[\sum_{j=1}^{n}\mathfrak{m}^2(0)^2\phi(\zeta + M_j^1(x;0))^2\right]^{1/2}\|\Delta\mathfrak{m}^2(t)\|_4\|\Delta M^1(x;t)\|_4$$

$$+ C_\phi^2\|\mathfrak{m}^2(0)\|\|\Delta\mathfrak{m}^2(t)\|_4\|\Delta M^1(x;t)\|_8^2 = O(\sqrt{n})$$

and

$$\left| \sum_{j=1}^{n} \Delta \mathfrak{m}_j^2(t)^2 \Delta \xi(M_j^1(t)) \right|$$

$$\leq 2C_\phi \mathbb{E}_{\zeta \sim \mathcal{N}(0,1)} \left[ \|\phi(\zeta + M^1(x;0))\|^2 \right]^{1/2} \|\Delta \mathfrak{m}^2(t)\|_8^2 \|\Delta M^1(x;t)\|_4 + C_\phi^2 \|\Delta \mathfrak{m}^2(t)\|_8^2 \|\Delta M^1(x;t)\|_8^2 = O(\sqrt{n}) \,.$$

With analogous reasoning, we can obtain that

$$\Delta \left( \frac{1}{n} \sum_{j=1}^{n} (\mathfrak{m}_j^2(t))^2 \psi(M_j^1(x;t))^2 \right) = O(1/\sqrt{n}) \,,$$

and so conclude that

$$\Delta Q^2(x;t) = O(1/\sqrt{n}) \,,$$

namely the output's variance is constant to the deterministic value $\sigma^2$ throughout the training, as $n \to \infty$.