# OpenReview forum: "A note on regularised NTK dynamics with an application to PAC-Bayesian training"
_TMLR — Accepted by TMLR_

### Review · Reviewer_Gxk2 · 2024-01-16

**Summary Of Contributions:**

The authors consider the question of how neural networks evolve in an infinite-width setting and extend current results from the neural tangent kernel (NTK) literature to regularized objectives, as long as these regularize the parameters to remain close to their initialization.

In the second part of their manuscript, they apply these results to PAC-based learning objectives, extending results by Clerico et al. (2023).

**Audience:**

Yes

**Claims And Evidence:**

Yes

**Requested Changes:**

None

**Strengths And Weaknesses:**

## Strengths
- The problem is well-motivated and focuses on a task that is relevant to the TMLR audience
- The paper is well organized and follows a clear structure.

## Weaknesses
- The paper is rather dense with respect to its notation, although that probably can't be avoided.
- Question: I am probably overlooking something, but where does the regularizing $\lambda$ appear from in (9)?

### Minor
- The manuscript seems at certain places unnecessarily hesitant/humble for the results the authors discuss. E.g., in the abstract "potentially contribute", or in the conclusion "appears to be a regularized version".
- Minor notation. It is somewhat overlapping that F is both the output as well as the Frobenius norm

### Typos
- Equation above (9) should be $\hat \ell$ instead of $\ell$ unless I am mistaken
- Sec 5, 2nd line: $\hat \ell(\hat y, y) = \frac{1}{2}(\hat y - y)^2$

---

> ### Author Response · Authors · 2024-02-18
> **Authors' reply**
>
> We thank the reviewer for their detailed and constructive feedback. We aim to improve our work with their suggestions.
>
> We thank the reviewer for pointing out a few typos, that we will fix for the final version of the paper.
>
> As for eq 9, the factor $\lambda$ is due to the definition of $\mathcal{C}_s = \mathcal{L}_s + \lambda \mathcal{R}$, which leads to $\partial_t \mathcal{R} = \nabla_W \mathcal{R}\cdot \partial_t W  = -  \nabla_W \mathcal{R}  \cdot\nabla_W \mathcal{L}_s - \lambda \|\nabla_W\mathcal{R}\|^2$. For $\mathcal{R} = \|\Delta W\|^2/2$, the last term becomes $-\lambda \|\Delta W\|^2 = -2\lambda\mathcal{R}$.
>
> As suggested we will try to change the tone of the paper and make it less hesitant/humble. We will also find an alternative for the Frobenius norm.

---

> > ### Comment · Reviewer_Gxk2 · 2024-02-19
> >
> > Thank you for your reply and clarification.

---

### Review · Reviewer_phPi · 2024-01-17

**Summary Of Contributions:**

This works extends the framework of neural tangent kernel to the case when regularization is used (though a special type), and applies the theory to understand PAC bayes learning.

**Audience:**

Yes

**Claims And Evidence:**

Yes

**Requested Changes:**

I think the authors should add the following discussion to the paper. Essentially, the result of the paper can be understood through a simple Taylor expansion.

In the NTK regime, the model is linear around the initialization:
$$f(\theta) = f(\theta_0) + \nabla^T f(\theta_0) \Delta \theta$$

This means that the loss function is also essentially quadratic in $\theta$.

But the regularization term under consideration is also quadratic in $\theta$. Therefore, the result studied by the authors can be simply understood by observing that the effective loss function is nothing but:
$$L = L0 +   V^T \Delta \theta + (\Delta \theta)^T (H + \lambda)\Delta \theta,$$
where $\lambda$ is the regularization strength, and $V$, $H$ is a constant vector/matrix.

No wonder the resulting dynamics is still linear. Adding this discussion to the paper can make the results of the paper much easier to understand for ordinary readers and improves the paper in my opinion.

**Strengths And Weaknesses:**

A main strength is that there is no way to study regularization in the framework of NTK yet. While this work does not make too much a progress, it is a decent progress forward I think. (Also, see the requested change section for why I think the progress is limited)

In terms of weakness, I do not have much to complain about. I think the theory is straightforward, but this is not a problem.

Given the novelty of the paper and there is no major concerns in its results I do lean towards acceptance

---

> ### Author Response · Authors · 2024-02-18
> **Author's reply**
>
> We thank the reviewer for their detailed and constructive feedback. We aim to improve our work with their suggestions.
>
> We thank the reviewer for suggesting to include in the paper a more intuitive explanation of the linearised regularised dynamics, which we believe will indeed make the main message of the work more clear.

---

### Review · Reviewer_qo7i · 2024-01-28

**Summary Of Contributions:**

The paper analyzes the infinite width (lazy; Neural Tangent Kernel (NTK) regime) training dynamics with a regularizer equal to [a monotonic function $\rho$ of] the $\ell^2$ distance of weights to their initialization (not to $0$). The paper:

* proves that in the infinite width limit the NTK stays constant throughout such regularized training;

* derives closed-form dynamics for $\rho \equiv 1$ (i.e. $\ell^2$ weight decay to initialization) and shows that it's equivalent to regularizing the network outputs (to initialization);

* shows that a certain setting of PAC-Bayesian training in the infinite width limit yields similar training dynamics (NTK with a regularization to init term).

**Audience:**

Yes

**Broader Impact Concerns:**

No ethical issues -- a theory paper.

**Claims And Evidence:**

Yes

**Requested Changes:**

## Major:

* Please outline novelty / impact of your results relative to https://arxiv.org/abs/1905.11368

* Could you elaborate on the importance / impact of section 6, describe specific use-cases / follow-ups to it?

  * (Overall, the paper would benefit from explaining its impact / importance more)

## Minor:

* Under equation 3: would be good to add a brief comment on what $\Xi$ represents.

* Section 4.1, "Evolution of the training objective": when reading the paper linearly the purpose of this subsection isn't clear, would be good to motivate it / put it more into context.

* Section 4.2 mentions a "... general regularizing term which still leads to tractable training dynamics". Could you elaborate in what way is it tractable? Aren't equations in Theorem 2 not solvable in closed form?

* I assume $F(x; t)$ in section 5 (immediately before section 6) can be solved in closed form? If so, would be good to mention it / write down the solution.

* Some light proof-reading could be useful, e.g.

	* Abstract: "yields an additional term [which?] appears"
    * Intro: "NTK dynamics prevent~~s~~"
    * Intro: "approach ~~approach~~"
    * Section 2: "Data consists ~~in~~[of]"
    * Before equation 5: "~~Moreover,~~[Where]"

**Strengths And Weaknesses:**

## Strengths:

* Overall the paper is well-written and easy to read (modulo minor issues listed in requested changes).

* Connection to PAC-Bayesian training appears interesting, but I'm not well-familiar with this field to properly understand its importance.

## Weaknesses:

* The case of $\rho \equiv 1$ (i.e. $\ell^2$ weight decay to initialization) seems to have been considered already in https://arxiv.org/abs/1905.11368 (Theorem 4.1).

    * Impact and/or motivation for generalizing to other monotonic $\rho$ isn't obvious -- the paper doesn't consider examples of other $\rho$, and from my understanding the dynamics in this case do not admit closed-form solutions.

* Implications from the connection to PAC-Bayesian training are also not very clear, and would benefit if motivation / envisioned consequences were more specific and spelled-out.

* The paper has no experiments, and to my understanding no straightforward/immediate applications.

For the above reasons I am on the fence regarding whether the paper is of high-enough interest to the TMLR audience.

---

> ### Author Response · Authors · 2024-02-18
> **Author's reply**
>
> We thank the reviewer for their detailed and constructive feedback. We aim to improve our work with their suggestions.
>
> We thank the reviewer for pointing out the work https://arxiv.org/abs/1905.11368, which we were not aware of and we believe it is worth mentioning in our paper. However, the Theorem 4.1 therein seems to not prove the convergence to linear dynamics in the limit of infinite width, as in the statement they take as an assumptions that the linearised dynamics holds (in the assumptions they use the model with linearised loss $\tilde L_\lambda$ and not the original one of eq 3 therein). We hence believe that our work can still be of interest in the case $\rho\equiv 1$, as we do provide a rigorous proof of why the linearised dynamics works.
>
> We will provide more context on the PAC-Bayesian training. Several recent works (see e.g. [1,2,3]) have studied the training of over-parameterised networks via training a generalisation (PAC-Bayes) objective. This allows to automatically get a generalisation guarantee for the trained model. Although these works have achieved promising empirical results, we are not aware of analytical studies of their training dynamics. Having a close form expression for this type of training could for instance help studying the convergence of GD (rates of convergence, convergence to global/local minima, etc.).
>
> In the context of PAC-Bayesian training, the training objective is usually non-linear in the KL divergence, and hence cannot be studied as a simple $\ell^2$ regularisation. However, the result of Theorem 2, with a general $\rho$, allows to study more general PAC-Bayesian training objectives, as the often used bound from [4] (see for instance [1]) or tightest variants such as [5] (see [3]). We will make sure to expand on this on the final version of our paper, in order to better justify the study of the general $\rho$ case. Although these dynamics cannot be solved exactly, we believe that having a closed-form ODE can still be of interest and potentially help the analytical study of the convergence of GD in such cases.
>
> [1] G. K. Dziugaite and D. M. Roy. Computing nonvacuous generalization bounds for deep (stochastic) neural networks with many more parameters than training data. UAI, 2017.
> [2] M. Pérez-Ortiz, O. Risvaplata, J. Shawe-Taylor, and C. Szepesvári. Tighter risk certificates for neural networks. JMLR, 22, 2021.
> [3] E. Clerico et al., Conditionally Gaussian PAC-Bayes, AISTATS, 2022.
> [4] D. A. McAllester. PAC-Bayesian model averaging. COLT, 1999.
> [5] A. Maurer. A note on the PAC Bayesian theorem. arXiv:0411099, 2004.

---

### Decision · Action_Editor_4jre · 2024-03-05

**Recommendation:** Accept with minor revision

**Comment:**

The reviewers have made a number of concrete suggestions for improvement but the paper has not been revised yet to incorporate them. Therefore, a revision is required as a condition for the paper's full acceptance. Specifically, I would like to see the following incorporated in a revision:

- All comments made by Reviewer qo7i under "Requested changes" (both "Major" and "Minor").
- All comments made by Reviewer phPi under "Requested changes".
- All comments made by Reviewer Gxk2 under "Minor" and "Typos".

**Audience:**

The paper extends the theory of the Neural Tangent Kernel to include regularized dynamics, and discusses the extended theory's application to PAC-Bayesian training. While the reviewers point out that these results might be of relatively low novelty or significance, they all agree that the results are of interest to the relevant community, so the paper meets TMLR's criteria.

**Claims And Evidence:**

The paper contains theoretical claims and no empirical results. All three reviewers agree that the theoretical claims are rigorously proven, so the paper clearly meets TMLR's evaluation criteria.